# Effect of a Multi-Strain Probiotic on Growth and Time to Reach Full Feeds in Preterm Neonates

**DOI:** 10.3390/nu14214658

**Published:** 2022-11-03

**Authors:** Marwyn Sowden, Evette van Niekerk, Andre Nyandwe Hamama Bulabula, Jos Twisk, Mirjam Maria van Weissenbruch

**Affiliations:** 1Division of Human Nutrition, Department of Global Health, Faculty of Medicine and Health Sciences, Stellenbosch University, Cape Town 8000, South Africa; 2Infection Control Africa Network—ICAN, Cape Town 7530, South Africa; 3Department of Epidemiology and Data Science, Amsterdam UMC, 1081 HV Amsterdam, The Netherlands; 4Amsterdam UMC, Department of Pediatrics-Neonatology, Vrije Universiteit Amsterdam, 1081 HV Amsterdam, The Netherlands

**Keywords:** growth, preterm neonate, probiotic, feeding intolerance

## Abstract

Background: The main nutritional goal for premature neonates is to achieve a postnatal growth rate that the neonate would have experienced in utero. Postnatal growth failure is, however, very common in very and extremely low birth weight neonates. The use of probiotics shows promising results in reducing the time for full feeds, as well as in increased weight gain. The optimal probiotic strain has, however, not been elucidated. The aim of the present study was to evaluate the difference in the growth and time to reach full feeds between the two treatment arms, using Labinic^TM^ as a multi-strain probiotic and a placebo. Methods: We conducted a double-blind, placebo-controlled, randomized clinical trial investigating the effect of a multi strain probiotic (Labinic^TM^) on various outcomes in preterm neonates. The results on the time to reach full feeds and the growth will be discussed in this paper. A probiotic or placebo was given once daily to the neonates for 28 days. Weight and feeding volume were measured daily, and length and head circumference were measured weekly. Results: The probiotic group reached full feeds earlier 8.7 days; ± 2.0 than the placebo group 9.7 days; ±4.3 (*p* = 0.04) and regained their birthweight earlier than the placebo group 11.5 days ± 6.3 vs. 13.3 days ± 6.3 (*p* = 0.06). From day 21 onwards, the probiotic group showed a significantly greater crude gain in weight (*p* < 0.001) than the placebo group (estimated difference between the two groups day 21: 56.7 g and at day 28: 83.7 g. There was a significant improvement observed in the weight Z-score change in the probiotic group over the 28-day period. Conclusion: The use of a multi-strain probiotic (Labinic^TM^) shows great potential as a low-cost, low-risk intervention in reducing the time to reach full feeds as well as shortening the time to regain birthweight. The probiotic had an additional beneficial impact on Z-score change in weight potentially decreasing post-natal growth restriction.

## 1. Introduction

The main nutritional goal for premature neonates is to achieve a postnatal growth rate that the neonate would have experienced in utero. This is unfortunately not easily achieved, and many premature neonates experience nutritional deficiencies that affect their weight, length, and head circumference [1,2].

Postnatal growth failure is very common in very and extremely low birth weight neonates [3]. Simmer found a positive correlation between the increased risks for long-term growth failure and a low gestational age and birth weight [4]. The National Institute of Child and Human Development indicated that 89–99% of premature neonates show growth failure at 36 weeks corrected age [5]. Long-term growth restriction and poor neurodevelopmental outcomes are associated with growth failure in these preterm neonates. On the other hand, postnatal growth failure, followed by catch up growth, is associated with a risk factor for metabolic syndrome [4,6,7,8]

The use of probiotics in premature neonates shows promising results. A systematic review and meta-analysis by Athalye-Jape et al. showed that 19 out of 25 trials indicated that the use of probiotics can reduce the time to full enteral feeds, with fewer episodes of feeding intolerance and improved weight gain and growth velocity (defined as weight gain per day) [6]. Additionally, a decreased transition time from orogastric to breast feeds, increased postprandial mesenteric flow, and a reduced hospital stay were observed [6]. Neonates supplemented with probiotics (irrespective of *Bifidobacterium* or non-*Bifidobacterium* strains; single or multiple strains or early and late initiation of probiotics) took less time to achieve full feeds, compared to the placebo groups. The optimal probiotic strains have not yet been elucidated. The aims of the present study were (i) to evaluate the difference in time to reach full feeds and (ii) to compare growth between the two treatment arms, using Labinic^TM^ as a multi-strain probiotic.

## 2. Materials and Methods

### 2.1. Study Design

We here present the results of a double-blind, placebo-controlled, randomized clinical trial investigating the effect of a multi strain probiotic on the time to reach full feeds and on growth in preterm neonates. This was one of the aims of a larger study, with the main aim of investigating the effect of probiotics on the carriage rate of DR-ESBL in preterm neonates [9].

### 2.2. Study Setting

The study was conducted at Tygerberg Hospital (TBH), Cape Town, South Africa. Participants were recruited, and data were collected over a period of 6 months, from 19 January to 27 June 2021.

### 2.3. Study Participants

Male and female preterm neonates, with a birth weight between 750–1500 g and a gestational age <37 weeks, were recruited. Neonates with major congenital malformations, early onset sepsis (C-reactive protein (CRP) >10 mg/L in the first 72 h of life), preterm neonates up for adoption, major gastro-intestinal abnormalities, or surgery of the gastro-intestinal tract were excluded.

### 2.4. Randomization

Neonates were recruited and randomized within the first 72 h after birth. We randomly allocated neonates to the two study arms—a probiotic (intervention) group (n = 100) and a placebo group (n = 100)—with the use of a pre-determined randomization list, obtained from a statistician at the University of Stellenbosch (Appendix A). The researcher and all neonatology staff were blinded as to which of the two groups received the probiotic versus placebo. Preterm neonates meeting the inclusion criteria were selected to be consecutively sampled, until the required sample size groups were achieved.

### 2.5. Procedures

#### 2.5.1. Probiotic/Placebo

The probiotic used was Labinic^TM^ (Biofloratech, Surrey, UK), and the placebo consisted of medium chain triglyceride (MCT) oil and Aerosil 200 (Aerosil 200 is a stabilizer used in Labinic^TM^, as well). Labinic^TM^ consists of *Lactobacillus acidophilus* (0.67 billion colony forming units (CFU)s), *Bifidobacterium bifidum* (0.67 CFUs), and *Bifidobacterium infantis* (0.67 CFUs).

The standard dose of 0.2 ml was administered once daily for 28 days, providing 2 billion CFUs per day. Supplementation with the probiotic or placebo was delayed if the neonate was nil per os (NPO) and discontinued if a neonate developed necrotizing enterocolitis (NEC) (Bells stage II or more). The researcher added the probiotic/placebo to neonate’s feed (mother’s own breast milk/donor breast milk/infant formula) before administration of the feed via an orogastric tube or, if applicable, orally. Neonates were followed up from birth to a maximum of 28 days/death/discharge to peripheral hospitals or home, whichever time point came first.

#### 2.5.2. Demographics and Medical Records

Data collected at enrollment included neonates estimated gestational age (early/late ultrasound or fetal foot length as recorded in the file), gender, birth weight, type of delivery, ethnicity, and Apgar scores. Medication prescribed, clinical, and laboratory notes were collected and documented daily. Neonates were screened daily by the attending neonatologist for the development of feeding intolerances. As per TBH neonatal protocol, nasogastric residuals were not measured. For the purpose of this study, feeding intolerance was identified when abdominal distension and/or emesis was encountered and led to a disruption of the feeding plan. The color, volume, and frequency of vomits and open nasogastric drainage were noted, as well as abdominal distension, stool volume, and consistency [10]. We also documented HIV exposure, this means the newborn was exposed to HIV virus (maternal HIV infection), but the HIV status of the newborn was not yet known (positive or negative).

#### 2.5.3. Anthropometry

Weight was measured daily, except when deemed unsafe by the attending neonatologist. Daily weights were recorded from each participant’s medical file by the investigator. If there were any drastic weight changes, the weight measurement was repeated by the investigator. Weight was measured on an electric scale with an accuracy of 1 g. Weight was measured to the nearest gram. Length and head circumference were measured at birth, on days 7, 14, 21, and 28 of life, by the investigator. Standard, non-stretchable measuring tapes, with 0, 5, and 1 cm dimensions, were used to measure the head circumference. A measuring rod (Seca 207) was used to measure the length. Length and head circumference were measured to the nearest 0.5 cm [11,12].

#### 2.5.4. Nutrition

The standard feeding protocol for preterm born neonates, as per TBH, was followed. The type and volume of feeds was recorded daily from each participant’s file. Full feeds were defined at a volume of 160 mL/kg/day.

#### 2.5.5. Definitions Size at Birth

Birth weight was classified as small for gestational age (SGA—weight < 10th percentile for gestational age (z score < −1.28)), appropriate for gestational age (AGA—weight between 10th and 90th percentile for gestational age (z score between −1.28 and +1.28)), or large for gestational age (LGA—weight > 90th percentile for gestational age (z score > +1.28)) [13].

### 2.6. Statistical Analysis

The current study was a secondary outcome of the main study aiming to determine a reduction in carriage rate of antibiotic resistant organisms with the supplementation of a multi-strain probiotic. The total sample size of the main study was 200, with 100 neonates per group (probiotic and placebo groups). It was estimated using a published decrease (17%) in the proportion of rectal colonization with drug-resistant bacteria [14]. This sample size was estimated to detect a significant difference between the groups being compared (with Type I error at 0.05 and power at 80%). The total sample size required allowed for a 12% margin for study participant lost to follow-up.

To analyze the difference in time to reach full feeds between the two treatment arms, we used the definition of 160 mL/kg/day as full feeds. We calculate the average time the neonates took to reach this feeding volume.

To analyze the difference in growth (in weight, length, and head circumference) between the two treatment arms, linear mixed model analyses were used. First, the difference, on average, over time between the two groups was analyzed, and second, the difference between the groups at the different time points was analyzed. For the latter, time (treated as a categorical variable, represented by dummy variables) and the interaction between treatment and time were added to the mixed model analyses. In all analyses, an adjustment was made for the baseline value of the particular outcome. All neonates’ weight, length, and head circumference measurements from birth up to day 28/death/discharge was also analyzed using the Fenton growth chart (https://peditools.org/fenton2013/, accessed on 6 April 2022), in order to determine the growth in Z-scores (Fenton, et al., 2013 [13]).

For all statistical tests performed, a *p*-value < 0.05 was considered significant. All the statistical analyses were performed using STATA 16.0 (College Station, TX, USA).

### 2.7. Ethical Approval

Ethical approval was granted by the Human Research Ethics Committee of the Faculty of Health Sciences of Stellenbosch University (S20/07/178), as well as the permission of Tygerberg Academic Hospital. The trial was also registered at the Pan African Clinical Trial Registry (PACTR202011513390736). As per ethical guidelines, informed consent was obtained from all mothers.

## 3. Results

Table 1 describes the basic demographic information of the 200 neonates enrolled in the study. The majority of neonates were females, with a birth weight above 1001 g (mean birthweights of the probiotic and placebo groups were 1174 g ± 226 g and 1150 g ± 230 g, respectively). Most of the neonates were born between 29–32 weeks of gestation (mean gestational ages of the probiotic and placebo group 29 weeks ± 13.9 days and 30 weeks ± 13.5 days, respectively), and their gestational age sizes at birth were appropriate. Almost three-quarters of the neonates were born via caesarean section.

The initiation of enteral feeds in the probiotic group was not significantly different between the groups (*p* = 0.87; Table 2), although the probiotic group reached full feeds earlier than the placebo group (*p* = 0.04; Table 2). Almost all the neonates received breastmilk as their first feed, except for one neonate in the probiotic group, who received formula milk. Most of the neonates continued to receive breastmilk for more than 50% of the study period. The neonates in the probiotic group regained their birthweight by day 11.5, on average, compared to day 13.3 in the placebo group (*p* = 0.06; Table 3).

The mean weight at birth was slightly higher in the probiotic than in the placebo group (1176 ± 227 g vs. 1152 ± 231 g). The probiotic group showed a significant greater weight gain during the study period, compared to the placebo group (1324 ± 253 g vs. 1200 ± 259 g; *p* < 0.001).

The mean birth length was slightly higher in the probiotic than in the placebo group (37.7 ± 3.3 g vs. 37.6 ± 3.1 cm). The probiotic group showed a statistically significant greater length accretion than the placebo group (39.7 ± 3.1 vs. 38.7 ± 3.0 cm; *p* = 0.006).

The mean head circumference at birth was slightly higher in the probiotic than in the placebo group (27.4 ± 2.0 cm vs. 27.2 ± 2.1 cm). As seen for weight and length, a similar trend was found for the head circumference accretion in the probiotic group, compared to the placebo group (28.6 ± 2.0 cm vs. 28.0 ± 1.7 cm; *p* = 0.17).

Figure 1 shows the mean crude change in anthropometric measurements (weight, length, and head circumference) during the study period.

Table 4 indicates the crude weight, length, and head circumference changes over time and at the different time points. The change in weight over time was statistically significant in probiotic group, from day 21 onwards. The change in length was statistically significant only at day 28. There was no statistically significant change in the head circumference.

Figure 2 shows the Z-score change in anthropometric measurements (weight, length, and head circumference) during the study period.

Table 5 indicates the weight, length, and head circumference Z-score change over time and at the different time points. The Z-score change in weight was statistically significant in probiotic group from day 21 onwards, as well as over time.

There was no significant difference in Z-score change in either the length or head circumference between the two groups. 

## 4. Discussion

The study showed that the use of a multi-strain probiotic has a positive impact on the time to reach full feeds in preterm neonates. Although the initiation of enteral feeds was slightly later in the probiotic group, they managed to reach full feeds earlier than the placebo group and regained their birthweight earlier. During the study period, the probiotic group showed a significantly greater crude weight gain. There was also a significant improvement observed in the weight Z-score change in the probiotic group over the 28-day period.

Early initiation of enteral feeds has been found to have several benefits for the preterm neonates. Benefits include a reduction in the number of days taken to reach full feeds, improved feeding tolerance, enhanced micronutrient delivery, promotion of intestinal development, and maturation and stimulation of the gut microbiome. A reduction in inflammation and decreased length of stay can also be observed. The early initiation of enteral feeds can also be a potential modifiable risk factor for both NEC and late-onset sepsis (LOS) [15,16,17]. Long-term benefits include enhanced brain growth, as well as neurodevelopment, avoidance of short stature, and catch-up growth, initiating risk factors for metabolic disorders [6,17].

After the initiation of enteral feeds, the aim is to reach full enteral feeds as soon as possible. Previous studies indicated that probiotics have a positive role to play in this regard. Early supplementation of *Bifidobacterium bifidum* (within 24–48 h after birth) versus a placebo lead to improvement in time to reach a feeding volume of 100 mL/kg/d 11.0  ±  3.6 days vs. 12.1  ±  3.8 days, *p* < 0.05), as well as improved daily body weight gain [18,19].

Athalye-Jape analyzed data from 19 trials and concluded that neonates supplemented with probiotics reached full feeds quicker, irrespective of whether *Bifidobacterium* stains, non-*Bifidobacterium*, or single strains were used (Athalye-Jape, et al., 2014). We found that the use of a multi-strain probiotic (*Lactobacillus acidophilus*, *Bifidobacterium bifidum*, and *Bifidobacterium infantis)* also led to an improvement in time to reach full feeds of 160 mL/kg/day (8.7 ± 1.99 days vs. 9.7 ± 4.31 days), without any adverse events.

Athalye-Jape also compared the use of a multi-strain probiotic with a single strain probiotic, showing that the first can reduce time to full enteral feeds by 1.74 days, whereas a single strain probiotic only reduced the time by 1.34 days [6]. Methods by which probiotics can increase feeding tolerances include increased gut maturity and gut motility by increased intestinal transit time, increased gastric emptying, and increased superior mesenteric artery flow [6]. Bifidobacteria form the major strain of the intestinal flora in healthy newborns. These enteric colonies assist with the normal development of the intestine by modulating the intestinal mucosal immunity (enhanced immunoglobulin A response), regulating the systemic immune response, producing anti-inflammatory cytokines, and the competitive inhibition of pathogenic bacteria. Bifidobacteria further aids in the digestion of proteins and carbohydrates, the synthesis of vitamins, short-chain fatty acids, and bacteriocins, and the maturation and differentiation of the intestinal mucosa [20]. The microbiome of the premature neonate is often dominated by pro-inflammatory γ-Proteobacteria that contain TLR4 ligands. It is thought that this increase in γ-Proteobacteria is actually a signature dysbiosis characteristic in the preterm. It would, thus, be ideal if a probiotic can reduce the number of γ-Proteobacteria and subsequently reduce inflammation in the neonate gut by reducing the TLR4 activity [21].

Preterm neonates are known to be at risk for growth failure with a lower weight and length and a suboptimal body composition (increased body fat percentage and a reduced lean mass), compared to their term equivalent age neonate [22]. Except for an optimal nutritional intake, there is, however, another key factor that needs to be considered in premature neonate growth—IGF-1 levels. Yumani et al. concluded that low early postnatal IGF-1 levels seem to be at the origin of growth failure, and that an increase in dietary protein (supplementation) can increase growth and improve body composition in premature neonates [23]. Moyer-Mileur compared four studies that examined the number of days to regain birth weight in premature neonates. Their results were that preterm neonates took approximately 0–17 days to regain their birth weight. On average, our probiotic group regained their birth weight by day 11.5, and the placebo group regained their birth weight by day 13.3 [24]. Furthermore, the probiotic group showed a statistically significant crude weight gain, compared to the placebo group. There was also an improved change in the Z-score for weight in the probiotic group. Yamasaki et al. used *B. bifidum* and concluded that daily weight gain was significantly higher in the supplemented group that the placebo group (mean weight gain of 21.4, compared to 18.3 g) [19]. Another study that used *B. lactis*, *B. longum*, or a combination of *B. lactis* and *B. longum*, however, showed no significant difference in growth, compared to a placebo [25]. These findings might highlight the point that different probiotic strains have different effects on preterm neonates, especially on growth.

A limitation of this study was the high proportion of the study population that was transferred out to peripheral hospitals, owing to high occupancy rates at the tertiary hospital, which led to reduced days of observation during the trial.

## 5. Conclusions

Our study showed that the use of a multi-strain probiotic has a positive effect on the time to reach full feeds. The probiotic had an improvement on the Z-score change in weight, potentially decreasing post-natal growth restriction.

The full trial protocol is available from the main author.

## Figures and Tables

**Figure 1 nutrients-14-04658-f001:**
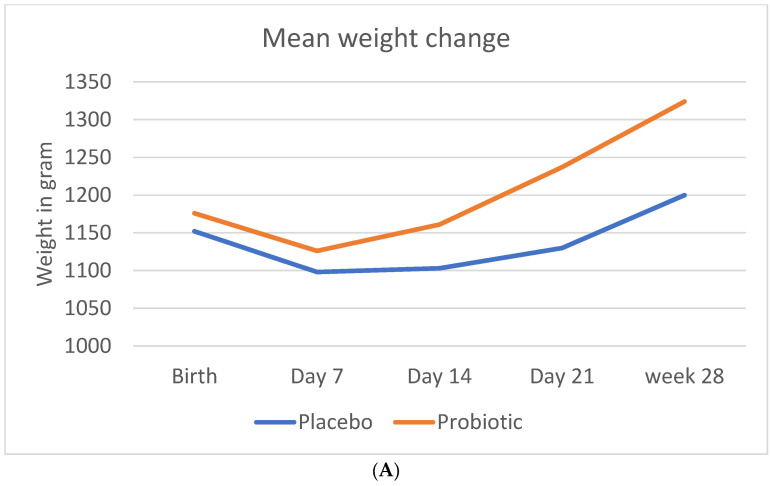
**Change in anthropometric measurements during the study period.** (**A**) Mean crude change in weight (**B**) Mean crude change in length (**C**) Mean crude change in head circumference.

**Figure 2 nutrients-14-04658-f002:**
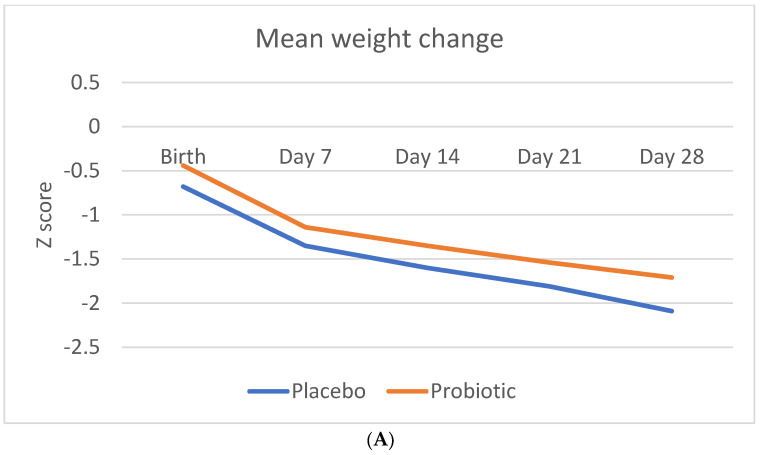
**Change in anthropometric measurement Z-scores during the study period**. (**A**) Mean Z-score change in weight (**B**). Mean Z-score change in length (**C**). Mean Z-score change in head circumference.

**Table 1 nutrients-14-04658-t001:** Neonatal data of the neonates enrolled in the study.

	Probiotic Group (n = 100)	Placebo Group (n = 100)
**Gender**		
Male (n, %)	47 (47)	37 (37)
Female (n, %)	53 (53)	63 (63)
**Birth weight**		
750–1000 g (n, %)	30 (30)	32 (32)
1001–1500 g (n, %)	70 (70)	68 (68)
**Gestational age**		
26–28 weeks (n, %)	34 (34)	30 (30)
29–32 weeks (n, %)	60 (60)	62 (62)
33–36 weeks (n, %)	6 (6)	8 (8)
**HIV**		
Exposed (n, %)	22 (22)	26 (26)
Unexposed (n, %)	78 (78)	74 (74)
**Mode of delivery**		
C-section (n, %)	73 (73)	73 (73)
Vaginal delivery (n, %)	27 (27)	27 (27)
**Birth number**		
Single neonate (n, %)	79 (79)	86 (86)
Twin neonates (n, %)	21 (21)	14 (14)

**Table 2 nutrients-14-04658-t002:** Nutritional intake of the neonates enrolled in the study.

	Probiotic Group n = 100	Placebo Group n = 100
Time of initiating enteral feeds (DOL mean, SD, and range)	3.1 ± 1.1 (0–6)	3.0 ± 1.0 (2–6)
Days to reach full feeds of 160 mL/kg/day (DOL mean, SD, and range)	8.7± 2.0 (5–18)	9.7 ± 4.3 (6–28)
**First feed received**		
EBM (n, %)	68 (68)	69 (69)
DEBM (n, %)	12 (12)	6 (6)
PEBM (n, %)	19 (19)	25 (25)
FM (n, %)	1 (1)	0 (0)
**Subsequent feeds received: ***		
EBM (n, %)	63 (63)	66 (66)
DEBM (n, %)	13 (13)	9 (9)
PEBM (n, %)	15 (15)	24 (24)
FM (n, %)	9 (9)	1 (1)
**Feeds fortified:**		
FM85	76 (76)	80 (80)
MCT oil	3 (3)	2 (2)

DEBM: Donor expressed breastmilk; DOL: Day of life; EBM: Expressed breastmilk; FM: Formula milk; FM 85: human milk fortifier; MCT oil: medium-chain triglyceride oil; PEBM: Pasteurized expressed breastmilk. * The feed received most often (>50% of the time).

**Table 3 nutrients-14-04658-t003:** Weight data of the neonates enrolled in the study.

	Probiotic Group n = 100	Placebo Group n = 100
**Size at birth**		
SGA (n, %)	17 (17)	23 (23)
AGA (n, %)	80 (80)	74 (74)
LGA (n, %)	3 (3)	3 (3)
**Types of growth restriction in SGA neonates**		
Symmetrical	8 (8)	9 (9)
Asymmetrical	9 (9)	14 (14)
**Weight**		
Birthweight in grams (mean, SD)	1174 g; ±226 g	1150 g; ±230 g
Days to regain birthweight (DOL mean, SD and range)	11.5 ± 6.3 (1 to 28)	13.3 ± 6.3 (4 to >28)

SGA: Small for gestation age; AGA: Appropriate for gestational age; LGA: Large for gestational age; DOL: Day of Life.

**Table 4 nutrients-14-04658-t004:** Change in the crude anthropometric measurements of the neonates over the trial period.

	Estimated Difference	95% CI	*p*-Value
**Weight**			
On average over time	33.7	11.0 to 56.4	0.004 *
Day 7	2.9	−20.1 to 25.9	0.81
Day 14	18.8	−6.2 to 43.8	0.14
Day 21	56.7	29.3 to 84.0	<0.001 *
Day 28	83.7	54.3 to 113.2	<0.001 *
**Length**			
On average over time	0.1	−0.1 to 0.2	0.33
Day 7	0.0	−0.1 to 0.1	0.83
Day 14	−0.0	−0.1 to 0.1	0.96
Day 21	0.1	−0.0 to 0.2	0.10
Day 28	0.2	0.1 to 0.3	0.006 *
**Head circumference**			
On average over time	0.1	−0.2 to 0.4	0.46
Day 7	−0.4	−0.8 to 0.1	0.13
Day 14	0.3	−0.2 to 0.8	0.25
Day 21	0.3	−0.2 to 0.9	0.25
Day 28	0.4	−0.2 to 1.1	0.17

* *p* < 0.05.

**Table 5 nutrients-14-04658-t005:** Change in the anthropometric measurement Z-scores of the neonates over the trial period.

	Estimated Difference	95% CI	*p*-Value
**Weight**			
On average over time	0.08	0.01 to 0.16	0.03 *
Day 7	0.01	−0.07 to 0.08	0.88
Day 14	0.05	−0.03 to 0.13	0.23
Day 21	0.12	0.03 to 0.21	0.007 *
Day 28	0.22	0.13 to 0.32	<0.001 *
**Length**			
On average over time	0.08	−0.09 to 0.25	0.35
Day 7	−0.08	−0.28 to 0.13	0.45
Day 14	0.17	−0.05 to 0.40	0.13
Day 21	0.16	−0.09 to 0.41	0.20
Day 28	0.19	−0.08 to 0.46	0.16
**Head circumference**			
On average over time	0.00	0.16 to 0.16	0.98
Day 7	−0.02	−0.18 to 0.14	0.78
Day 14	−0.05	−0.22 to 0.11	0.51
Day 21	0.03	−0.14 to 0.20	0.70
Day 28	0.09	−0.09 to 0.26	0.33

* *p* < 0.05.

## Data Availability

Data supporting reported results can be requested from the main author.

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
