# Peer review of "Effect of a Multi-Strain Probiotic on Growth and Time to Reach Full Feeds in Preterm Neonates"

_nutrients, 2022, doi:10.3390/nu14214658_

Round 1
Reviewer 1 Report
line 100: "the probiotc/placebo...:" is redundant, already described in line 95.
line 104: why estimated gestational age and not calculated from LMP? also explain the role of late ultrasound for calculation of gestational age.
line 109: how feeding intolerance was evaluated. please explain. also no data about possible feeding intolerance are reported, and this is very common in VLBW and ELBW neonates. this is a maior issue.
line 163 and line 165: it is not useful to give both birth weight and weeks as, respectively, above 1001g and between 29-32. instead range and mean, as in table, should be reported.
in table 1 exposition to HIV is reported but this is not discussed anywhere. which exposition is? what exposition means? there is non difference but it could be useful some explanation of this very high report.
line 189, 195 and 200 insert p=NS.
line 285 improved Z-score for head circumference is reported but this is not so in table 5. maior issue
the limitation reported in line 292 needs to be discussed, because in secondary hospital or home (as written in line 102) doubts about supplementation and measurements arise (if it is correct my intrerpretation that during the period of 28 days of the study newborns can be allocated elsewhere). maior issue
line 298: again an improvement of Z-score is reported.
Author Response
Dear Reviewer,
Thank you for reviewing the article, below please find the correspondence.
line 100: "the probiotc/placebo...:" is redundant, already described in line 95. Line 110. Thank you for the suggestion, duplication was removed.
line 104: why estimated gestational age and not calculated from LMP? also explain the role of late ultrasound for calculation of gestational age. Line 114. Thank you for the comment, since this is a tertiary hospital, the hospital often receives patients from outlying areas with minimal obstetric care. Mothers are often seen for the first time just before or during labour. The gestational age and how it was calculated was recorded from the neonate’s file.
line 109: how feeding intolerance was evaluated. please explain. also no data about possible feeding intolerance are reported, and this is very common in VLBW and ELBW neonates. this is a maior issue. Line 117-121. This is described in a previous article published by us, and references. A brief description was provided in this article.
line 163 and line 165: it is not useful to give both birth weight and weeks as, respectively, above 1001g and between 29-32. instead range and mean, as in table, should be reported. Line 180-184. Thank you for the comment. The range and means are provided in the text. The table provides the distributions for birthweight categories and gestational age week categories.
in table 1 exposition to HIV is reported but this is not discussed anywhere. which exposition is? what exposition means? there is non difference but it could be useful some explanation of this very high report. Line 122-124. HIV exposure was added to the methodology. The HIV exposure was almost the same between the two groups. Statistics in South Africa show that around 30% of pregnant woman have HIV, and thus give born to a HIV-exposed infant.
line 189, 195 and 200 insert p=NS. Line 192 and Line 197. Thank you for the suggestion. We prefer to give the actual p-value instead of NS. We believe that it is important, because for instance there is a remarkable difference between p=0.06 and p=0.87.
line 285 improved Z-score for head circumference is reported but this is not so in table 5. maior issue Line 330. Thank you, removed the head circumference as it was not significant.
the limitation reported in line 292 needs to be discussed, because in secondary hospital or home (as written in line 102) doubts about supplementation and measurements arise (if it is correct my intrerpretation that during the period of 28 days of the study newborns can be allocated elsewhere). maior issue. Thank you for the comment, as soon as the neonates left Tygerberg Hospital, follow-up of the neonates was discontinued. Logistically and financially, we were not able to follow these neonates up at peripheral hospitals/home.
line 298: again an improvement of Z-score is reported. Line 34. Thank you for the comment, we removed the head circumference as it was not significant.
Kind regards,
Marwyn

Reviewer 2 Report
This study showed that the use of a multi-strain probiotic has a positive effect on time to reach full feeds.
The probiotic administered (Labinic) within 2-3 days od delivery during 1 month, a combination of 1 strain of Lactobacillus and 2 strains of Bifidobacterium has proved to be very effective and free of side-effects.
had an improvement on Z-score change in weight and head circumference, potentially decreasing post-natal growth restriction.
--
Q 1. Title :
Is very informative, complete and has a clear message
Q 2. Abstract and Keywords :
The main nutritional goal for premature neonates is to achieve a postnatal growth rate that the neonate would have experienced in utero. Postnatal growth failure is, however, very common in very and extremely low birth weight neonates. The use of probiotics shows promising results in reducing the time to full feeds as well as in increased weight gain. The optimal probiotic strain has, however, not been elucidated. The aim of the present study was to evaluate the difference in growth and time to reach full feeds between the two treatment arms, using LabinicTM as a multi-strain probiotic and a placebo.
Methods: We conducted a double-blind, placebo-controlled, randomized clinical trial investigating the effect of a multi strain probiotic (LabinicTM) on various outcomes in preterm neonates. The results on the time to reach full feeds and the growth will be discussed in this paper. A probiotic or placebo was given once daily to the neonates, for 28 days. Weight and feeding volume were measured daily, length and head circumference were measured weekly. Results: The probiotic group reached full feeds earlier 8.7 days; ± 2.0 than the placebo group 9.7 days; ±4.3 (p=0.04) and regained their birthweight earlier than the placebo group 11.5 days 25 ± 6.3 versus 13.3 days ± 6.3 (p=0.06). From day 21 onwards, the probiotic group showed a significantly greater crude gain in weight (p < 0.001), than the placebo group (estimated difference be- tween the two groups day 21: 56.7 grams (g) and at day 28: 83.7 g).
Conclusion: The use of a multi-strain probiotic (LabinicTM) shows great potential as a low-cost, low risk intervention in reducing the time to reach full feeds as well as shortening the time to regain birth-weight. The probiotic had an additional beneficial impact on Z-score change in weight potentially decreasing post-natal growth restriction.
The Keywords are well selected, informative and in a Good number
Q 3. 1. Introduction :
Postnatal growth failure is very common in very- and extremely low birth weight neonates (Dusick, et al., 2003). Simmer found a positive correlation between the increased risks for long-term growth failure and a low gestational age and birth weight (Simmer,2007). The National Institute of Child and Human Development indicates that 89% - 99% of premature neonates show growth failure at 36 weeks corrected age
Long-term growth restriction and poor neurodevelopmental outcomes are associated with growth failure in these preterm neonates. On the other hand, postnatal growth failure followed by catch up growth are associated with a risk factor for metabolic syndrome
The use of probiotics in premature neonates shows promising results. A systematic review and meta-analysis by Athalye-Jape et al showed that 19 out of 25 trials indicated that the use of probiotics can reduce the time to full enteral feeds, fewer episodes of feeding intolerance, improved weight gain and growth velocity (defined as weight gain per day) (Athalye-Jape, et al., 2014). Additionally, a decreased transition time from orogastric to breast feeds, increased postprandial mesenteric flow and a reduced hospital stay was observed
Neonates supplemented with probiotics (irrespective of Bifidobacterium or non-Bifidobacterium strains; single or multiple strains or early and late initiation of probiotics) took less time to achieve full feeds compared to the placebo groups (Athalye-Jape, et al., 2014). The optimal probiotic strains have yet not been elucidated. The aims of the present study were (i) to evaluate the difference in time to reach full feeds and (ii) to compare growth between the two treatment arms, using LabinicTM as a multi-strain probiotic.
Q 4. 2. Materials and Methods
Study design
We here present the results of a double-blind, placebo-controlled, randomized clinical trial investigating the effect of a multi strain probiotic on the time to reach full feeds and on growth in preterm neonates. This was one of the aims of a larger study, with the main aim of investigating the effect of probiotics on the carriage rate of DR-ESBL in preterm neonates. (Under press).
Study setting
The study was conducted in Tygerberg Hospital (TBH), Cape Town, South Africa Participants were recruited, and data collected over a period of 6 months, from 19 January to 27 June 2021.
Study participants
Male and female preterm neonates, with a birth weight between 750 – 1500 g and a gestational age <37 weeks were recruited. Neonates with major congenital malformations, early onset sepsis (C-reactive protein (CRP) >10mg/L in the first 72 hours of life), preterm neonates up for adoption, neonates with major gastro-intestinal abnormalities or surgery of the gastro-intestinal tract were excluded.
Randomization
Neonates were recruited and randomized within the first 72 hours after birth. We randomly allocated neonates to the two study arms – a probiotic (intervention) group (n=100) and a placebo group (n=100) with the use of a pre-determined randomization list, obtained from a statistician at the University of Stellenbosch. The researcher and all neonatology staff were blinded as to which of the two groups received the probiotic versus placebo. Preterm neonates meeting the inclusion criteria was selected to be consecutive sampled until the required sample size groups were achieved.
Procedures
Probiotic/placebo:
The probiotic used was LabinicTM (Biofloratech, Surrey, United Kingdom) and the placebo consisted of medium chain triglyceride (MCT) oil and Aerosil 200 (Aerosil 200 is a stabilizer used in LabinicTM as well). LabinicTM consists of Lactobacillus acidophilus (0.67 92 billion colony forming units (CFU)s), Bifidobacterium bifidum (0.67 CFUs) and Bifidobacterium infantis (0.67 CFUs).
The standard dose of 0.2ml was administered once daily for 28 days, providing 2 billion CFUs per day. Supplementation with the probiotic or placebo was delayed if the neonate was nil per os (NPO) and discontinued if a neonate developed necrotizing enterocolitis (NEC) (Bells stage II or more). The researcher added the probiotic/placebo to neonate’s feed (mother’s own breast milk / donor breast milk / infant formula) before administration of the feed via an orogastric tube or if applicable, orally. The probiotic/placebo was administered once daily for 28 days and neonates were followed up from birth to a maximum of 28 days/death/discharge to peripheral hospitals or home, whichever time point came first.
Demographics and medical records:
Data collected at enrolment included neonates estimated gestational age (early/late ultrasound or fetal foot length), gender, birth weight, type of delivery, ethnicity, Apgar scores. Medication prescribed, clinical and laboratory notes were collected and documented daily. Neonates were screened daily by the attending neonatologist for the development of feeding intolerances (Sowden, et al., 2022).
Anthropometry:
Weight was measured daily except when deemed unsafe by the attending neonatologist. Daily weights were recorded from each participant’s medical file by the investigator. If there were any drastic weight changes, the weight measurement was repeated by the investigator. Weight was measured on an electric scale with an accuracy of 1 g. Weight was measured to the nearest gram. Length and head circumference were measured at birth, on day 7, 14, 21 and 28 of life by the investigator. A standard, non-stretchable meas- uring tape with 0,5 and 1cm dimensions were used to measure the head circumference. A measuring rod (Seca 207) was used to measure the length. Length and head circumference were measured to the nearest 0.5 cm (WHO, 2008), (Lee DL, 2003).
Nutrition:
The standard feeding protocol for preterm born neonates as per TBH was followed. The type and volume of feeds was recorded daily from each participant’s file. Full feeds were defined at a volume of 160 ml/kg/day.
Definitions size at birth:
Birth weight was classified as small for gestational age (SGA - weight < 10th percentile for gestational age (z score < -1.28)), appropriate for gestational age (AGA – weight between 10th and 90th percentile for gestational age (z score between -1.28 and +1.28)) or large for gestational age (LGA – weight > 90th percentile for gestational age (z score > +1.28) (Fenton, et al., 2013).
Statistical analysis
The current study was a secondary outcome of the main study aiming to determine a reduction in carriage rate of antibiotic resistant organisms with the supplementation of a multi-strain probiotic. The total sample size of the main study was 200, with 100 neonates per group (probiotic and placebo groups). It was estimated using a published decrease (17%) in the proportion of rectal colonization with drug-resistant bacteria (Xin- Tian, et al., 2014). This sample size was estimated to detect a significant difference between the groups being compared (with Type I error at 0.05 and power at 80%). The total sample size required was allowing a 12% margin for study participant lost-to-follow-up.
To analyze the difference in time to reach full feeds between the two treatment arms, we used the definition of 160ml/kg/day as full feeds. We calculate the average time the neonates took to reach this feeding volume.
To analyze the difference in growth (in weight, length, and head circumference) between the two treatment arms, linear mixed model analyses were used. First, the difference on average over time between the two groups was analyzed and second, the difference between the groups at the different time points was analyzed. For the latter, time (treated as a categorical variable, represented by dummy variables) and the interaction between treatment and time were added to the mixed model analyses. In all analyses an adjustment was made for the baseline value of the particular outcome. All neonates’ weight, length, and head circumference measurements from birth up to day 28/death/discharge was also analyzed using the Fenton growth chart (https://peditools.org/fen- 150 ton2013/) in order to determine the growth in Z-scores (Fenton, et al., 2013).
For all statistical tests performed, a p-value < 0.05 was considered significant. All the statistical analyses were performed using STATA 16.0 (College Station, Texas 77,845 153 USA).
Ethical approval
Ethical approval was granted by the Human Research Ethics Committee of the Faculty of Health Sciences of Stellenbosch University (S20/07/178) as well as a permission from Tygerberg Academic Hospital. The trial was also registered at the Pan African Clinical Trial Registry (PACTR202011513390736). As per ethical guidelines, informed consent was obtained from all mothers.
Q 5. 3. Results
Table 1 described the basic demographic information of the 200 neonates enrolled in the study. The majority of neonates were females, with a birth weight above 1001 g (mean birthweight of the probiotic and placebo group 1174 g ± 226 g and 1150 g ± 230 g respectively). Most of the neonates were born between 29-32 weeks of gestation (mean gestational age of the probiotic and placebo group 29 weeks ± 13.9 days and 30 weeks ± 13.5 days, respectively) and their gestational age size at birth was appropriate. quarters of the neonates was born via caesarean section.
The initiation of enteral feeds in the probiotic group was not significantly different between the groups (p=0.87; table 2), although the probiotic group reached full feeds earlier than the placebo group (p=0.04; table 2). Almost all the neonates received breastmilk as their first feed, except for one neonate in the probiotic group, who received formula milk. Most of the neonates continued to receive breastmilk more than 50% of the study period. The neonates in the probiotic group regained their birthweight on average by day 11.5 compared to day 13.3 in the placebo group (p=0.06; table 3).
The mean weight at birth was slightly higher in the probiotic tan in the plkacebo group (1176 ± 231g). The probiotic group showed a significantly greater weight gain during the study period compared to the placebo group (1324 ± 253 g. versus 1200 ± 259 g, p<0.001)
The mean birth length was slightly higher in the probiotic than in the placebo group (37.7 ± 3.3 gram versus 37.6 ± 3.1 cm). The probiotic group showed a statistically significant greater length accretion than the placebo group (39.7 ± 3.1 versus 38.7 ± 3.0 cm; p = 0.006)
The mean head circumference at birth was slightly higher in the probiotic than in the placebo group (27.4 ± 2.0 cm versus 27.2 ± 2.1 cm). As seen for weight and length, a similar trend was found for the head circumference accretion in the probiotic group compared to the placebo group, (28.6 ± 2.0 cm versus 28.0 ± 1.7 cm; p = 0.17).
Table 4 indicates the crude weight, length and head circumference change over time and at the different time points. The change in weight over time was statistically significant in probiotic group, from day 21 onwards. The change in length was statistically significant only at day 28. There was no statistically significant change in the head circumference.
Table 5 indicates the weight, length and head circumference Z-score change over time and at the different time points. The Z-score change in weight was statistically significant in probiotic group from day 21 onwards as well as over time.
There was no significant difference in Z-score change in either length or head circumference between the two groups.
Q 6. 4. Discussion
Early initiation of enteral feeds has been found to have several benefits for the preterm neonates. Benefits include a reduction in the number of days taken to reach full feeds, improved feeding tolerance, enhanced micronutrient delivery, promotion of intestinal development as well as maturation and stimulation of the gut microbiome. A reduction in inflammation, and decreased length of stay can also be observed. Early initiation of enteral feeds can also be a potential modifiable risk factor for both NEC and late-onset sepsis.
Long term benefits include enhanced brain growth as well as neurodevelopment, avoidance of short stature and catch-up growth initiating risk factors for metabolic disorders
After the initiation of enteral feeds, the aim is to reach full enteral feeds as soon as possible. Previous studies indicated that probiotics have a positive role to play in this regard. Early supplementation of Bifidobacterium bifidum (within 24-48h after birth) versus a placebo lead to improvement in time to reach a feeding volume of 100ml/kg/d (11.0 ± 3.6 days versus 12.1 ± 3.8 days, p < 0.05), as well as improved daily body weight gain
Athalye-Jape analyzed data from 19 trials and concluded that neonates supplemented with probiotics reached full feeds quicker, irrespective if Bifidobacterium stains, non-Bifidobacterium or single strains were used (Athalye-Jape, et al., 2014). We found that the use of a multi-strain probiotic (Lactobacillus acidophilus, Bifidobacterium bifidum and Bifidobacterium infantis) also leads to an improvement in time to reach full feeds of 160ml/kg/day , without any adverse effects
Methods by which probiotics can increase feeding tolerances, include increased gut maturity and gut motility by increased intestinal transit time, increased gastric emptying and increased superior mesenteric artery flow (Athalye-Jape, et al., 2014). Bifidobacteria form the major strain of the intestinal flora in healthy newborns. These enteric colonies assist with the normal development of the intestine by modulating the intestinal mucosal immunity (enhanced immunoglobulin A response), regulating the systemic immune response, production of anti-inflammatory cytokines, competitive inhibition of pathogenic bacteria.
Bifidobacteria further aids in the digestion of proteins and carbohydrates, synthesis of vitamins, short-chain fatty acids and bacteriocins, and the maturation and differentiation of the intestinal mucosa (Lee, et al., 2005). The microbiome of the premature neonate is often dominated by pro-inflammatory γ-Proteobacteria that contain TLR4 ligands. It is thought that this increase in γ-Proteobacteria is actually a signature dysbiosis characteristic in the preterm. It would thus be ideal if a probiotic can reduce the number of γ-Proteobacteria and subsequently reduce inflammation in the neonate gut by reducing TLR4 activity
Preterm neonates are known to be at risk for growth failure with a lower weight and length and a suboptimal body composition (increased body fat percentage and a reduced lean mass) compared to their term equivalent age neonate (Johnson, et al., 2011). Except for an optimal nutritional intake there is however another key factor that needs to be considered in premature neonate growth – IGF-1 levels. Yumani et al concluded that low early postnatal IGF-1 level seem to be at the origin of growth failure, and that an increase in dietary protein (supplementation) can increase growth and improve body composition in premature neonates (Yumani, et al., 2015). Moyer-Mileur compared four studies which examined the number of days to regain birth weight in premature neonates. Their results were that preterm neonates took approximately 0–17 days to regain their birth weight.
On average our probiotic group regained their birth weight by day 11.5 and the placebo group by day 13.3 (Moyer-Mileur, 2007). Furthermore, the probiotic group showed a statistically significant crude weight gain compared to the placebo group. There was also an improved change in the Z-score for weight and head circumference in the probiotic group. Yamasaki et al used B. bifidum and concluded that daily weight gain was significantly higher in the supplemented group that the placebo group (mean weight gain of 21.4 compared to 18.3 grams) (Yamasaki, et al., 2012).
Another study that used B. lactis, B. longum or a combination of B. lactis and B. longum however showed no significant difference in growth compared to a placebo (Hays, et al., 2016). These findings might highlight the point that different probiotic strains have different effects in preterm neonates especially on growth.
A limitation of this study was the high proportion of the study population that was transferred out to peripheral hospitals, owing to high occupancy rates at the tertiary hospital, which led to reduced days of observation during the trial.
5. Conclusions
This study showed that the use of a multi-strain probiotic has a positive effect on time to reach full feeds. The probiotic had an improvement on Z-score change in weight and head circumference, potentially decreasing post-natal growth restriction.
References :
They have been well selected, recently published and very appopriated

Author Response
Dear Reviewer,
Thank you for reviewing the article. I could not find any suggestions/questions.
Kindly find attached the new draft with comments incorporated from the other reviewer.
Kind regards,
Marwyn
